# The First Assessments of Pediatric HBV Immunization Coverage in Mauritania and Persistence of Antibody Titers Post Infant Immunizations

**DOI:** 10.3390/vaccines11030588

**Published:** 2023-03-03

**Authors:** Hala El Hachimi, Mohamed Mahmoud Mohamed El Alem, Esma Haimoudane, Cheikh Yebouk, Jannie Pedersen, F-Zahra Fall-Malick, Fatimetou Khiddi, Mohamed Abdawe, Sidi Ahmed Sadegh, Hugues Fausther-Bovendo, Mohamed Vall Mohamed Abdellahi

**Affiliations:** 1Unit of Epidemiology Molecular and Diversity of Microorganisms, University of Nouakchott, Nouakchott 2373, Mauritania; 2National Reference Laboratory of Leishmaniasis, National Institute of Hygiene, Rabat 10010, Morocco; 3Laboratoire National de Contrôle de la Qualité des Médicaments, Nouakchott 5347, Mauritania; 4Global Urgent and Advanced Research and Development, Batiscan, QC G0X 1A0, Canada; 5Unit of Research: Marine Ecobiology, Environment, Health and Nutrition EBIOMESN, University of Nouakchott, Nouakchot 2373, Mauritania

**Keywords:** hepatitis B virus, seroprotection, humoral response, vaccine, children, Mauritania

## Abstract

Background: The Hepatitis B virus (HBV) vaccine is used worldwide as an efficient tool to prevent the occurrence of chronic HBV infection and the subsequent liver disease. However, despite decades of vaccination campaigns, millions of new infections are still reported every year. Here, we aimed to assess the nationwide HBV vaccination coverage in Mauritania as well as the presence of protective levels of the antibodies against HBV surface antigen (HBsAb) following vaccination in a sample of children immunized as infants. Methods: To evaluate the frequency of fully vaccinated and seroprotected children in Mauritania, a prospective serological study was conducted in the capital. First, we evaluated the pediatric HBV vaccine coverage in Mauritania between 2015 and 2020. Then, we examined the level of antibodies against HBV surface antigen (HBsAb) in 185 fully vaccinated children (aged 9 months to 12 years) by ELISA using the VIDAS hepatitis panel for Minividas (Biomerieux). These vaccinated children were sampled in 2014 or 2021. Results: In Mauritania, between 2016 and 2019, more than 85% of children received the complete HBV vaccine regimen. While 93% of immunized children between 0 and 23 months displayed HBsAb titer >10 IU/L, the frequency of children with similar titers decreased to 63, 58 and 29% in children aged between 24–47, 48–59 and 60–144 months, respectively. Conclusions: A marked reduction in the frequency of HBsAb titer was observed with time, indicating that HBsAb titer usefulness as marker of protection is short lived and prompting the need for more accurate biomarkers predictive of long-term protection.

## 1. Introduction

Hepatitis B virus (HBV) infection is a major public health problem worldwide and its infection causes more than one million deaths each year [1]. It is estimated that one-third of the world’s population has already been infected by this virus [2]. In highly endemic areas, particularly in parts of Africa and Southeast Asia, approximately 7–20% of people are chronically infected and more than 70% of adults have evidence of previous infection [3]. HBV may be acute or chronic, and may range from asymptomatic infection or mild illness to, in rare cases, more severe complications such as fulminant hepatitis, hepatocarcinoma, cirrhosis and HBV-related kidney disease [4,5]. 

The likelihood of chronic HBV infection is inversely proportional to the age of acquisition of infection [6]. Chronic HBV infection occurs in approximately 80% to 90% of perinatally infected children, 30% to 50% of children infected before the age of 6 years and less than 5% of infections in adults [4,7,8]. Thus, protection of young children against HBV infection is paramount, as they are highly vulnerable to disease complications [9,10]. Various therapeutic countermeasures against HBV are available for treatment-eligible children including interferon-based therapy, and nucleoside and nucleotide analogs. The antivirals entecavir and tenofovir, two nucleoside analogs with a high genetic barrier to resistance, are approved for the treatment of HBV-infected children aged 2–17 and 12–18 years, respectively. Both drugs aim to reduce viremia, and therefore, the risk of progression to hepatocellular carcinoma and cirrhosis. However, access to both drugs is extremely limited in low- and middle-income countries. Of note, interferon-alpha-based treatment, and its pegylated version, is approved for the treatment of children older than 1 and 2 years old in the US and Europe. However, due to their associated toxicity, high cost and need for frequent subcutaneous injections, interferon-alpha-based treatments, are not recommended for HBV-infected children in low- and middle-income countries [5]. In 2016, an estimated 10% of individuals with chronic HBV infection were aware of their HBV status. Furthermore, only 5% of treatment-eligible individuals were actually treated against HBV infection. Due to poor treatment availability and limited diagnostics, HBV prevention remains the best approach [11]. Vertical transmission from mother-to-child during delivery is considered the most frequent HBV transmission route to children, while postpartum and intrauterine transmissions have also been reported. Horizontal transmission between family members for children under 5 years of age is also common, especially in endemic areas [12,13]. The risk of vertical transmission increases with higher maternal HBV DNA levels and with vaginal delivery, even when passive or active at-birth prophylaxis is provided to the mother [14]. Indeed, in Asia, the risk of HBV mother-to-child transmission ranges from 70 to 90% and from 10 to 40% for HBeAg-positive and HBeAg-negative mothers, respectively. For yet undetermined reasons, the risk of mother-to-child transmission is lower in Africa. Management of HBV carriers and HBV vaccination are important in endemic areas [5].

Primary prevention through vaccination remains the most effective way to control the spread of HBV, particularly in developing countries [15]. Effective HBV vaccines were developed in the early 1980s. After three doses of vaccine given during the first 6 months of life, 90–99% of healthy neonates, children, adolescents and adults developed protective levels of antibodies against HBV surface antigen (HBsAb), with serum levels above the 10 IU/L threshold considered protective [16]. Based on the 2009 World Health Organization (WHO)’s recommendations, all infants, irrespective of maternal HBV serostatus, should receive 3–4 doses of HBV vaccine starting preferably within 24 h after birth to prevent HBV mother-to-child transmission [17]. If properly implemented, the above vaccine regimen alone or in combination with HBV immunoglobulin (HBIg) can reduce mother-to-child transmission of HBV by more than 90% [13]. The importance of the HBV vaccine is illustrated by the WHO recommendation to introduce it into the national immunization programs of all endemic countries. The current coverage of HBV vaccines is relatively high. In 2015, close to 80% of individuals in Africa were estimated to have received the required three doses of vaccines [18]. However, vaccination coverage is not evenly distributed throughout Africa. Notably, in the Central African Republic and Nigeria only 54 and 45% of children, respectively, received the complete three dose HBV vaccine regimen before turning 1 year old. Interestingly, approximately 7.1% of global HBV infection occurs in Nigeria. This country is ranked third in terms of global HBV infection, behind China and India, which account for an estimated 29.5 and 11.2% of global infection, respectively [11]. However, vaccination rate is an imperfect marker of vaccination efficiency. Although booster vaccination is not recommended by the WHO, many countries around the world have observed a drop in HBsAb levels over a period of 3–6 years to levels <10 IU/L [19]. Furthermore, a small but significant proportion of children do not mount an antibody response of sufficient magnitude after three doses of hepatitis B vaccination [19]. Documenting the level of HBsAb over time in fully immunized children is therefore paramount to improve the current vaccination schedules against HBV.

This study, which was conducted in Mauritania, documents the nationwide proportion of fully vaccinated children as well as the presence of protective levels of HBsAb in toddlers which received the complete three dose vaccine regimen against HBV before the inclusion of the hepatitis B vaccine dose at birth.

## 2. Materials and Methods

### 2.1. Evaluation of Vaccine Coverage

We sought to document the proportion of toddlers which were fully protected by the existing HBV vaccine in Mauritania. The national vaccination coverage in Mauritania was evaluated. To that end, the number of toddlers under one year old in Mauritania, and the three districts of its capital Nouakchott, were estimated based on information provided by the Mauritanian national office of statistics. The number of fully vaccinated toddlers in the corresponding areas were also gathered from the national expanded child immunization program. Toddlers who received a prime followed by 2 boosts of an HBV protein subunit vaccine containing the HBV surface antigen (HBsAg) were considered fully vaccinated. It is worth mentioning that the estimate of number of toddlers under one year old was not always accurate. Notably, in Nouakchott West, the number of fully vaccinated toddlers was higher than the estimate number of toddlers under one year old, resulting in erroneous vaccination coverage above 100%.

Data on immunization coverage were obtained from the Mauritanian expanded child immunization program (Programme élargie de vaccination). The local national office of statistics provided the information related to population size in Mauritania between 2015 to 2020.

### 2.2. Study Population

Altogether, one hundred eighty-five children were enrolled in this study. Children aged between 9 months and 6 years, were recruited at the Centre Hospitalier Mère Enfant (CHME) in Nouakchott between July and December 2014. A second group of schoolchildren (6–12 years) were sampled in Nouakchott from January to May 2021.

Prior to sampling, a brief interview was conducted to obtain informed consent from the guardians and completion of an information sheet. Only children who received at least 3 doses of the hepatitis B vaccine were included in this study. This subunit vaccine, denoted Hepatitis B vaccine recombinant DNA (cat# S359323) which contains the HBsAg was purchased from the Serum Institute of India (Pune, India). Exclusion criteria included any acute infectious disease (measles, rubella, hepatitis A and hepatitis B) within the previous two weeks; a blood transfusion within six months prior to the study; immunosuppressive state due to any diseases or the use of immunosuppressive therapy.

### 2.3. Sampling

For each child that met the inclusion criteria, a 3 mL venous blood sample was collected using a vacutainer-type sampling system, in a dry tube. All collected blood samples were centrifuged (1000 rpm at 4 °C) for 10 min. The recovered sera were kept at −20 °C for approximately 72 h prior to laboratory analysis.

### 2.4. Hepatitis B Virus Serology

To rule out active HBV infection, all 185 specimens were tested for the presence of HBV surface antigen (HBsAg). All samples were also screened for the presence of antibodies against HBV surface antigen (HBsAb). Both assays were performed using the VIDAS hepatitis panel using the Minividas system (Biomerieux, Craponne, France). HBsAb greater than 10 IU/L is considered protective [20]. This threshold was therefore used in our study.

### 2.5. Data Analysis

Data processing and analysis was performed using Graphpad Prism (version 9.4) and the python package: Seaborn Version: 0.11.2. A *p*-value <0.05 was considered statistically significant. One-way Annova (Kruskal–Wallis test) followed by the Dunn’s test was used to compare HBsAb titers between the different age groups.

## 3. Results

Protective HBV vaccine regimen are commercially available. To understand the cause of the elevated HBV prevalence in Mauritania, we first analyzed immunization coverage in local children. Local numbers of children under one year old and of fully vaccinate children were gathered from the Mauritanian national office of statistics and the national expanded child immunization program, respectively. Between 2015 and 2020, vaccination coverage in Mauritania was relatively high, ranging from 73 to 81% with peak vaccination coverage in 2017 and 2019. A similar trend was observed for the Nouakchott North and Nouakchott South districts of the capital, as well as the Nouakchott West district which recorded a high vaccination coverage exceeding 99% (Table 1). In 2020, vaccination coverage declined, including in the three districts of Nouakchott, as well as in the country as a whole. It is worth noting that in Nouakchott West the estimated HBV vaccination coverage was above 100%. This suggests that the population of children younger than one year old was significantly underestimated in this area. However, the trend in HBV vaccine coverage in Nouakchott West was similar to the ones observed in the whole country and in other districts of the capital.

Full vaccination does not always result in protective immunity. To better evaluate protection against HBV in children from Mauritania, we evaluated both the presence of active HBV infection and the humoral response generated in fully vaccinated children. To that end, children who received the recommended three doses of HBV vaccine were enrolled. Children were segregated based on age. A total of 42, 30, 17 and 96 children ages 0–23, 24–47, 48–59 and 60–144 months were recruited, respectively (Table 2). All children were tested for the presence of HBV surface antigen (HBsAg), which is indicative of an active infection; all the tested children were negative. The absence of active HBV infection in any of the tested children is another illustration of the protective efficacy of the current HBV vaccine. In the study cohort, 92.9% of immunized children aged 0–24 months had protective titer (>10 IU/L), confirming the ability of the current vaccine regimen to induce strong humoral response against HBV (Figure 1A). However, the proportion of fully vaccinated children with humoral response above 10 IU/L sharply drops from 92.9% in the 0–24 months, to 63.3, 58.8 and 29.2% in children aged 24–47, 48–60 and 61–144 months, respectively (Figure 1A).

In our cohort, the proportion of female children greatly varies between the various groups ranging from 29.4% in the 48–59 months group to 52.1% in the 60–144 months groups (Table 2). To ensure that the difference in gender was not responsible for the drop in humoral response against HBV surface antigen, our data was disaggregated by sex. The sharp decrease in HBsAb levels was observed over time in both male and female participants. Between the 0–23 and 60–144 months groups, average HBsAb titers declined from 233.5 to 32.0, from 205.2 to 29.6 and from 275.1 to 34.4 IU/L, when the data included both sexes, males and females, respectively. In all three cases, the decrease in the HBsAb titer was statistically significant with p values below 0.001. Of note, the decline in HBsAb titers between the 0–23, and 24–47 or 48–59 months groups were not always statistically significant (Figure 1B). This discrepancy was mainly driven by the low number of participants aged between 24 and 60 months (Table 2).

Overall, this study confirms that the current three dose HBV immunization schedule is protective and induces a strong humoral response in young children. However, humoral immunity rapidly wanes. In Mauritania, by the age of 2 years, roughly 60% of children retain a humoral response of magnitude above the 10 IU/L threshold. The frequency of children with a humoral response above this threshold, further drops to 30% in 5- to 12-year-old children. This decrease in humoral response was observed independently of the sex of the immunized children.

## 4. Discussion

In Mauritania, infectious diseases such as HBV are now considered a severe public health problem. Vaccination is considered the most cost-effective way to control HBV infection, and early vaccination of newborns is essential to prevent perinatal HBV transmission. The HBV vaccine is part of the WHO Expanded Programme on Immunization (EPI), illustrating the importance of this vaccine. This study aimed to evaluate the HBsAg seroconversion in children 9 months–12 years and the level of seroprotection against HBV, as defined by an HBsAb titer above 10 IU/L, in Mauritania. This is the first study to assess the impact of EPI after 15 or more years of introduction of the HBV vaccine in the EPI.

The successful introduction of the HBV vaccine into the National Expanded Immunization Program in Mauritania has a great impact on the prevalence of HBsAg in the population aged 9 months to 12 years. Following the introduction of HBV vaccination in Mauritania in 2004, there has been a noticeable decrease in HBV prevalence. Indeed, in a prospective survey in 2019 and 2020, the seroprevalence of HBV infection decreased significantly at3%, 0.7% and 0.5% for those aged ≤10–14, 0–4 and 5–9 years, respectively, while the prevalence reported in 1990 among children aged 0 to14 years exceeded 16% [21,22]. These findings illustrate the important impact of the integration of the HBV vaccine into the EPI in Mauritania and globally as similar results were observed in Senegal [23], and in Asian and Western countries [24].

In the present study, no active HBV infection was detected in fully immunized children, further demonstrating the potency of the HBV vaccine. The HBsAg prevalence observed in children from the capital Nouakchott was lower than in children from other part of Mauritania [21], in healthy Indonesian children (3.1%) [25] and in Senegalese children (1.23%) [26].The difference observed with other African countries is probably associated with several factors including different vaccination coverage and malnutrition.

In Mauritania, HBV vaccine coverage is relatively high, with more than 85% of children fully vaccinated between 2016 and 2019. However, HBV vaccine coverage decreased in 2020, probably due to the ongoing COVID-19 pandemic. Indeed, pediatric vaccination coverage against several infectious diseases including polio virus, tuberculosis, measles and rubella, declined during the COVID-19 pandemic [27]. For a long-lasting decrease in HBV prevalence in Mauritania, local efforts by the National Expanded Immunization Program in Mauritania should be sustained to keep elevated HBV immunization coverage in toddlers. The recent decline in HBV vaccine coverage that is probably due to the current coronavirus disease 2019 (COVID-19) should therefore be promptly addressed in order not to lose the progress achieved by the local immunization program.

Our study shows that the prevalence of HBsAb titer above 10 IU/L is 52.97%. Similar studies in different countries such as Bangladesh (50.2%) [19], Taiwan (35.9%), Saudi Arabia (38.0%), the United States of America (24.0%) and Slovakia (48.4%) also showed similar results [28].

Of note, as vaccine coverage is an imperfect marker of protection, the present study monitors the development of the antibody response against the HBV surface antigen. Shortly after vaccination, in the 0–23 months old group, all study participants seroconverted, with HBsAb titers above 3 IU/L. Similar high seroconversion levels were observed in immunized children from Bangladesh (99.9%) [29], Brazil (98%) [30], Sri Lanka (90.5%) [31] and Ghana (100%) [32].

In the 0–23 months old group, roughly 93% of vaccinated children developed HBsAb titers above the 10 IU/L protective threshold. In children 2–5 years old, only 60% of children had protective HBsAb titers. The proportion of children with titers above the protective threshold dropped further to 29% in 5- to 12-year-old children. While a limited number of participants of some of the age groups were available for this study, a similar decline in protective humoral response against HBV was reported different countries including Senegal [23], Burkina Faso [33], Bangladesh [19], South Korea [34], Yemen [35], China [36], Ghana [32], Palestine [37] and Egypt [38].

In absence of contradictory data on new HBV infections in immunized individuals, it is only presumed that the HBV vaccine confers long-term protection [39]. Despite a gradual decline in HBsAb titers post-HBV immunization, several 30-year-long follow-up studies have demonstrated that a three dose HBV vaccine regimen affords long-term protection against HBV in immunized neonates [40,41,42]. Several additional studies also demonstrated that HBV immunized individuals are able to generate robust cellular and humoral recall responses 5 to 23 years post-booster vaccination [39]. Altogether, the present study and others therefore suggest that despite decreasing levels of HBsAb, the HBV vaccine confers protection and therefore the use of the HBsAb titer above 10 IU/L as a marker of seroprotection is not appropriate in children older than 2 years old, and that individuals with HBsAb titer below 10 IU/L can still mount a robust recall response [40]. As a result, more accurate biomarkers of HBV protection are required.

While the HBV vaccine is highly efficacious, some breakthrough HBV infection can occur in immunized children, which can lead to various health complications such as primary liver cancer, severe end-stage liver diseases and infant fulminant hepatitis [40]. The identification of biomarkers capable of distinguishing immunized children susceptible to HBV infection is paramount. In these sub-groups of children, additional countermeasures, such as a fourth vaccine dose, might be necessary. It is worth pointing out that a booster dose of immunization is recommended in immunocompromised individuals, as it serves to activate the anamnestic response causing the HBsAb titers to go above the protective threshold value [8].

In addition to antibody-based biomarkers, more accurate biomarkers of HBV protection could be based on cellular or innate response. Cellular immunity is also believed to play an important role in HBV-vaccine-induced protection. Recent studies have shown protective T-cell responses decades after initial vaccination, even in patients with waning antibodies [43]. Even though the presence of HBV-specific T-cells might be a more sensitive indicator of protection, logistical constraints prevent its wider use in the clinic. In contrast, humoral immunity has been extensively studied over several decades and is still listed as an important factor in protection. The contribution of innate immunity to HBV-vaccine-induced protection has not been thoroughly studied. However, in clinical trials, superior humoral responses were observed in individuals immunized with CpG-adjuvanted HBV vaccine [44]. The yellow fever vaccine, one of the most effective vaccines, activated multiple toll-like receptors (TLRs) [45]. Overall, these studies suggest that the predictive value of innate-immunity-based biomarkers should be further evaluated in the context of HBV immunization.

More broadly, the in-depth study of the immune response generated following complete HBV immunization could help identify biomarkers of long-term vaccine-induced protection. The current coronavirus disease 2019 (COVID-19) has highlighted the need for such biomarkers [46]. Their identification could facilitate the selection of vaccine candidates, and reduce the length of time and cost of clinical evaluations.

## 5. Conclusions

Overall, the present study indicates HBV vaccination coverage in Mauritania is high and fully vaccinated children are protected from infection. Maintaining high vaccination coverage will be paramount to reducing HBV prevalence in Mauritania. Our study also suggests that the HBsAb titer is an imperfect marker of children protection against HBV. Indeed, by 2 years old, only 60% of vaccinated children have antibody level above the 10 IU/L titers and this frequency further declines overtime. As a result of decreasing levels of HBsAb titers, alternate markers of protective immunity are recommended. Additional studies focused on various immune markers, including but not limited to early inflammatory responses, in vitro B- and T-cell activity and anamnestic response are required in cohorts of immunized children to identify more accurate biomarkers of long-term HBV protection post-immunization.

## Figures and Tables

**Figure 1 vaccines-11-00588-f001:**
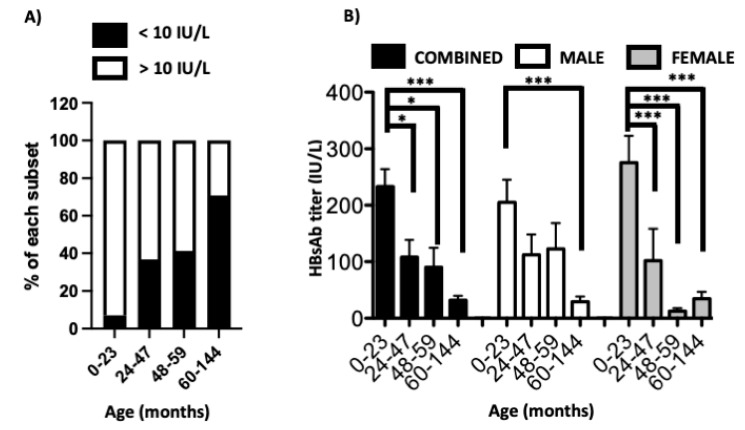
The proportion of children with significant antibody titers against HBV sharply decreases within years of immunization. (**A**). The proportion of children with humoral response above the 10 IU/L is illustrated for the four age groups. (**B**). The antibody level against HBV surface antigen [HBsAb] including both sexes (combined) and segregated by sex is indicated for the same age groups. * and *** represent *p* values below 0.05 and 0.001, respectively.

**Table 1 vaccines-11-00588-t001:** Three dose [HepB3] coverage. HBV vaccination coverage in toddlers younger than 1 year old was calculated for the districts of the capital Nouakchott and for the whole of Mauritania.

	2015	2016	2017	2018	2019	2020
Nouakchott North						
Total population	399,368	413,194	427,531	442,363	457,735	473,654
Estimated number of children < 1 year	14,377	14,875	15,391	15,925	16,478	17,052
Number of fully vaccinated children	10,064	11,751	12,621	11,785	13,018	11,595
Estimated proportion of fully immunized children	70%	79%	82%	74%	79%	68%
Nouakchott West						
Total population	180,482	186,730	193,209	199,912	206,859	214,053
Estimated number of children < 1 year	6497	6722	6956	7197	7447	7706
Number of fully vaccinated children	6722	8806	8347	8708	8341	7629
Estimated proportion of fully immunized children	103%	131%	120%	121%	112%	99%
Nouakchott South						
Total population	463,327	479,367	496,000	513,208	531,042	549,510
Estimated number of children < 1 year	16,680	17,257	17,856	18,475	19,118	19,782
Number of fully vaccinated children	11,342	11,735	12,677.755	12,748	13,573	12,265
Estimated proportion of fully immunized children	68%	68%	71%	69%	71%	62%
Mauritania						
Total population	3,720,125	3,805,659	3,893,775	3,984,233	4,077,347	4,173,077
Estimated number of children < 1 year	132,719	135,769	138,912	142,137	145,458	148,872
Number of fully vaccinated children	96,885	116,762	123,631	120,817	129,457	120,586
Estimated proportion of fully immunized children	73%	86%	89%	85%	89%	81%

**Table 2 vaccines-11-00588-t002:** Fully vaccinated children cohort. The number of samples as well as the frequency of female participants are indicated for each group.

Age Group [Months]	*N*, Frequency [%]	Sex [*n*, % of Female]
0–23	42 [22.7]	17 [40.5]
24–47	30 [16.2]	11 [36.7]
48–59	17 [9.2]	5 [29.4]
60–144	96 [51.9]	50 [52.1]
Total	185 [100]	83 [44.9]

## Data Availability

Not applicable.

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
