# Peer review of "The First Assessments of Pediatric HBV Immunization Coverage in Mauritania and Persistence of Antibody Titers Post Infant Immunizations"

_vaccines, 2023, doi:10.3390/vaccines11030588_

Round 1
Reviewer 1 Report
The study aimed to assess the nationwide HBV vaccination coverage in Mauritania as well as the presence of protective levels of the antibodies against HBV surface antigen (HBsAb) following vac-cination in a sample of children immunized as infants. The study evaluated the frequency of fully vaccinated and seroprotected children in Mauritania, using a prospective serological approach and was conducted in three districts of the capital Nouakchott using the VIDAS immunoassay. The study reported that more than 85% of children received the complete HBV vaccine regimens. While 93% of immunized children, between 0 and 23 months, displayed HBsAb titer >10IU/L, the frequency of children with similar titers decreased to 63, 58 and 29% in children aged between 24-47, 48-59 and 60-144months. The study showed marked reduction in HBsAb titer with time, indicating that HBsAb titer usefulness as marker of protection is short lived. The authors called for a need for more accurate biomarkers predictive of long-term protection.
The authors might consider modifying their call because of long-lasting immunity (even when titres fall below the accepted protective level of 10miu/mL) the need for these markers could be unnecessary. The authors could therefore place an emphasis on ensuring babies and children receive a complete and timely HB vaccine course.
Overall, the paper is well written and deserves publication.
Author Response
"Please see the attachment."

Reviewer 2 Report
The authors analyzed the pediatric HBV immunization coverage and the persistence of antibody titers in the infants. The information reported is important and interesting. Some aspects should be addressed before considering publishing this manuscript.
Major comments
1. Table 1: please explain why the % coverage are over 100 in Nouakchott West
- The rapid decrease in Ab levels in vaccinated children is surprising. Did the authors consider other factors, such as problems with the cold chain in previous years?
- The authors should compare their data with other reported studies, particularly the apparently very rapid decline in antibodies. Some studies have shown a rapid response to booster vaccination, suggestion the effective presence of memory B cells that could help to prompt a rapid response to an eventual exposure. This should be discussed.
- Even with low Ab levels, these children were not infected, particularly 0/96 after 5 years of vaccination, in agreement with the thought that the IU limit for considering protection against HBV infection may be in fact lower.
- The low sample number for some age groups may have hamper the comparison too.
Minor comments
6. Introduction page 2 line 50: the frequency of vertical transmission vs horizontal one depends on the geographic area, in endemic countries. Thus it would be advisable to add a some before endemic.
7. Page 2, line 54: why particularly in developing countries? Anywhere!
8. Page 2 line 64: the current coverage is not high in all developing countries, so please specify and add a reference.
9. Figure 1B: while Figure 1A is a frequency graph, 1B shows the average Ab levels, and the minimum accepted level (10IU/L) not shown. So this is not the same comparison.
Author Response
"Please see the attachment."
